# Comprehensive Analysis of the Prognosis and Drug Sensitivity of Differentiation-Related lncRNAs in Papillary Thyroid Cancer

**DOI:** 10.3390/cancers14051353

**Published:** 2022-03-07

**Authors:** Wenlong Wang, Ning Bai, Xinying Li

**Affiliations:** 1Thyroid Surgery Department, Xiangya Hospital, Central South University, Changsha 410008, China; 198102130@csu.edu.cn; 2National Clinical Research Center for Geriatric Disorders, Xiangya Hospital, Central South University, Changsha 410008, China

**Keywords:** papillary thyroid cancer (PTC), differentiation, risk score, drug sensitivity, long non-coding RNA

## Abstract

**Simple Summary:**

Merging evidence has indicated that dedifferentiation is the main concern associated with radioactive iodine (RAI) refractoriness in patients with papillary thyroid cancer (PTC), and the underlying mechanisms of PTC dedifferentiation remain unclear. This study systematically delineated the expression pattern, tumor immune microenvironment, drug sensitivity, and prognostic value of differentiation-related lncRNAs. It also demonstrated that DPH6-DT could be considered as a novel signature to indicate differentiation and promote TC progression via activating the *PI3K-AKT* signaling pathway.

**Abstract:**

Dedifferentiation is the main concern associated with radioactive iodine (RAI) refractoriness in patients with papillary thyroid cancer (PTC), and the underlying mechanisms of PTC dedifferentiation remain unclear. The present work aimed to identify a useful signature to indicate dedifferentiation and further explore its role in prognosis and susceptibility to chemotherapy drugs. A total of five prognostic-related DR-lncRNAs were selected to establish a prognostic-predicting model, and corresponding risk scores were closely associated with the infiltration of immune cells and immune checkpoint blockade. Moreover, we built an integrated nomogram based on DR-lncRNAs and age that showed a strong ability to predict the 3- and 5-year overall survival. Interestingly, drug sensitivity analysis revealed that the low-risk group was more sensitive to Bendamustine and TAS-6417 than the high-risk group. In addition, knockdown of DR-lncRNAs (*DPH6-DT*) strongly promoted cell proliferation, invasion, and migration via PI3K-AKT signal pathway in vitro. Furthermore, *DPH6-DT* downregulation also increased the expression of vimentin and N-cadherin during epithelial-mesenchymal transition. This study firstly confirms that DR-lncRNAs play a vital role in the prognosis and immune cells infiltration in patients with PTC, as well as a predictor of the drugs’ chemosensitivity. Based on our results, DR-lncRNAs can serve as a promising prognostic biomarkers and treatment targets.

## 1. Introduction

The incidence of thyroid cancer (TC) has been increasing in recent years and papillary TC (PTC) is the most frequent histological type that derives from follicular cells, accounting for up to 85% of cases [1]. Although the vast majority of patients with PTC have a favorable prognosis via reasonable treatments, including thyroidectomy, thyroid-stimulating hormone (TSH) suppressive therapy, and radioactive iodine (RAI) therapy, approximately 10–20% of PTCs suffer from disease recurrence and progress to distant metastasis during follow-up [2]. Among these patients, dedifferentiation is the main reason that leads to PTC transform into poorly differentiated or anaplastic TC (ATC) with poor clinical outcome. At present, the treatment options for these patients are limited, and the molecular mechanisms of PTC dedifferentiation remain unclear.

Recently, an increasing number of studies have reported that genetic and epigenetic aberrations [3,4], cancer stem cells [5], microRNAs [6], immunometabolic networks [7], and autophagy [8,9] play a critical role in PTC dedifferentiation and RAI resistance. Long non-coding RNAs (lncRNA), defined as a series of transcripts greater than 200 nucleotides, have been regarded as crucial regulators at various levels of transcriptional, post-transcriptional, and translational regulation. They are extensively involved in carcinogenesis, chromatin dynamics, interactions with mRNAs and proteins, differentiation, and embryonic development in patients with TC [10,11]. Notably, several studies proved that lncRNA could effectively modulate NF-κB and PI3K-AKT signaling pathways, thus affecting tumorigenesis [12,13], and lncRNACDC6 could serve as ceRNA to target CDC6 by sponging micro-225 to promote breast cancer progression [14]. At the same time, impairment or disturbances in lncRNA expression result in increased chemoresistance [15]. LncRNA CRNDE directly binds to SRSF6 and reduces the alternative splicing of PICALM, thereby mediating chemoresistance in gastric cancer [16]. Therefore, lncRNA can be considered a prognostic and therapeutic marker for cancer.

Some studies [17,18,19,20] revealed that lncRNA served as a potentially useful biomarker for the malignant thyroid nodule diagnoses, prognosis prediction, and treatment response of PTCs. Our previous study demonstrated that PTCSC3, as a tumor suppressor, was associated with prognosis in TC [19]. A systematic review and meta-analysis showed that lncRNA function as biomarker for TC diagnosis and lymph nodes metastasis prediction [21]. However, differentiation-related lncRNA (DR-lncRNA) mediating PTC dedifferentiation patterns is not yet fully elucidated.

Currently, tumor microenvironment (TME) has received extensive attention. Within the TME of TC, tumor cells interact with immune cell infiltration and coordinate the immune response that induces tumor dedifferentiation to promote PTC progression [22,23,24]. The proportions of immune cells, such as tumor-infiltrating lymphocytes, dendritic cells (DCs), and tumor-associated macrophages (TAMs), are closely related to the extent of differentiation [25,26]. Inhibition of intratumoral TAM recruitment might be able to restore RAI sensitivity in poorly differentiated TC [27]. In addition, immune checkpoint blockade (ICB) elicits durable and effective responses in solid tumors [28]. Anti-PD1/PD-L1 antibodies were effective in inhibiting ATC progression and improved survival dramatically [29]. Thus, better understanding the role of TME involved in PTC dedifferentiation and the development of new biomarkers to choose patients for ICB treatment is definitely required.

In this study, we performed a multi-step analysis to identify the prognostic significance of DR-lncRNAs based on The Cancer Genome Atlas (TCGA) and Gene Expression Omnibus (GEO) datasets and established an integrated nomogram to improve prognostic risk stratification. Furthermore, we explored the correlation between the corresponding risk score, TME, and drug sensitivity. More importantly, functional and molecular experiments were performed to validate the function of DR-lncRNAs. These results might be able to identify useful signatures which indicate dedifferentiation. These signatures may assist us in evaluating disease prognosis and therapeutic decisions for patients with PTC.

## 2. Materials and Methods

### 2.1. Subjects and Data Acquisition

We downloaded the RNA-Seq dataset and matched clinical information (including age, gender, histological type, bilaterality, multifocality, T stage, lymph node metastasis, distant metastasis, and survival time) from University of California Santa Cruz (UCSC) Xena, available online: http://xena.ucsc.edu/ (accessed on 2 October 2021). The samples with survival time less than 30 days or incomplete clinical data were excluded from the study. Finally, 492 TC and 58 normal samples from TCGA database were enrolled. Whole-transcriptome sequencing data was performed using FPKM expression level in transcripts per million (TPM). The median absolute deviation (MAD) was calculated to exclude genes with high variability [30]. LncRNAs with MAD > 0.5 were excluded from the RNA-Seq matrix. LncRNAs with an expression level of 0 or no clear name annotation were also excluded. In total, 1636 lncRNAs were enrolled.

According to published literature [31], there were 16 differentiation-related regulators, including *NKX2-1*, *DUOX1-2*, *PAX8*, *SLC5A5*, *SLC5A8*, *SLC26A4*, *FOXE1*, *TG*, *TSHR*, *THRA*, *THRB*, *DIO1-2*, *GLIS3*, and *TPO*. The microarray dataset GSE33630 comprising 11 anaplastic TC, 49 PTC, and 45 normal samples, was downloaded from GEO database. This dataset is based on the GPL570 (HG-U133_Plus_2) Affymetrix Human Genome U133 Plus 2.0 Array [32], was used to validate the differential expression of DR-lncRNAs. The workflow is shown in Figure 1.

### 2.2. Protein–Protein Interaction (PPI) Network

To analyze the potential relationship between differentiation-related gene regulators, the STRING database [33] was used to construct a PPI network and then screen out the hub genes. A minimum gene interaction score of 0.4 was set as a threshold for genes at the center of the PPI network.

### 2.3. Construction of the Prognostic Risk Assessment Model

First, Pearson’s correlation analysis was performed to filter the DR-lncRNAs based on the threshold criteria of Pearson’s coefficient |R^2^| > 0.5 and *p* < 0.001. Then, the least absolute shrinkage and selection operator (LASSO) Cox regression was used to select candidate prognostic DR-lncRNAs. The corresponding coefficient criteria and optimal penalty parameter lambda were determined through 10-fold cross-validation based on the minimum criteria. Subsequently, an ideal prognostic model was established. The risk score for each patient was calculated as follows: risk score = $\overset{n}{\underset{i=1}{\sum }}coef\left(i\right)*a\left(i\right)$, where *a(i)* represents the expression of DR-lncRNAs, and coef(i) is the coefficient. The patients were divided into high- and low-risk groups based on the median risk score.

### 2.4. GO and KEGG Pathway Enrichment Functional Analysis

To reveal the functions of differentially expressed genes (DEGs) in high- and low-risk score groups. Differentially expressed mRNAs were normalized and analyzed by using the “limma” package. *p* values < 0.05 and |log2FC| ≥ 1 were used as thresholds. GO and KEGG pathway enrichment analysis were visualized by Metascape [34] (http://metascape.org, accessed on 2 October 2021).

### 2.5. Evaluation of Immune Infiltration and the Expression of Immune Checkpoints

Cell type identification by estimating relative subsets of RNA transcripts (CIBERSORT) [35] was used to calculate the enrichment scores of the fraction of 22 immune cell types for each sample. The single-sample Gene Set Enrichment Analysis (ssGSEA) [36] was performed to quantify the enrichment level of 29 infiltrating immune cells, and MCP counter algorithms [37] were utilized to assess the proportion of immune cells. Furthermore, we compared the expression of immune checkpoint molecules (*PD-1, PD-L1, TNFSF9, IDO2, CD80*,* D44, CD27,* and *CD160*) in the low-risk score group and high-risk score group.

### 2.6. Drug Susceptibility Prediction

The CellMiner database [38] is based on the IC50 of over 20,000 compounds and 60 cancer cells listed by the National Cancer Institute’s Cancer Research Center (NCI-60). Drugs under clinical trials and FDA-approved drugs were obtained. Spearman correlation analysis was utilized to determine the relationship between differentiation-related genes and drug sensitivity, and a box plot was drawn. The threshold criteria of Pearson’s coefficient > 0.6 and *p* value < 0.01 were considered statistically significant.

### 2.7. Tissue, Cell Lines, and Cell Transfection

Twenty paired PTC samples and adjacent normal tissues were collected from patients who underwent thyroidectomy in the Thyroid Surgery Department of Xiangya hospital from March 2020 to June 2020, and then preserved in the refrigerator at −80 °C. Informed consent was obtained from all the participants and approved by the Ethics Committee of Xiangya Hospital of Central South University (No. 202004192). Human TC cells (BCPA-P, TPC-1, IHH-4, K-1, KTC-1, WRO, BHT-101, FRO, and 8305 C) and normal human thyroid cells (Nthy-ori-3-1) were purchased from the Tumor Cell Bank of the Chinese Academy of Medical Science. Cell were maintained in RPMI 1640 or high glucose DMEM with 10% fetal bovine serum (FBS) and 1% streptomycin/penicillin at 37 °C with 5% CO_2_.

Small interfering RNA (siRNA) targeting *DPH6-DT* was synthesized and designed by RiboBio company (Guanzhou, China). KTC-1 and IHH-4 cells were grown to 30% to 40% confluence in a 6-well plate and then transfected with *DPH6-DT* siRNA mixed with riboFECT^TM^CP (Wuhan, China) Buffer and reagent according to the manufacturer’s instruction.

### 2.8. RNA Extraction and Quantitative Reverse Transcription-PCR (qRT-PCR)

Total RNA was extracted from the tissue samples and cell lines by using the AG RNAex Pro RNA extraction kit (AG21101, Changsha, China). The first-strand cDNA was synthesized with the Reverse Transcription kit (AG, Changsha, China). qRT-PCR analysis was carried out using the TBGreen Premix Pro Taq HS Qpcr Kit (Cat #AG11718, Changsha, China) on ABI7500 system. The sequence of primers was listed in Appendix A. The relative expression levels of each sample were calculated using the ΔΔCq method, and the results were expressed as 2^−ΔΔCq^.

### 2.9. Western Blotting

Total proteins were extracted and separated by 10% SDS-polyacrylamide gel electrophoresis, and then transferred onto a nitrocellulose membrane (Millipore, Burlington, MA, USA). The membrane was incubated in 5% low-fat milk in Tris-buffered saline with 0.1% Tween-20. Subsequently, the membranes were incubated with the rabbit antibodies against human E-cadherin, N-cadherin, vimentin, p-AKT, AKT, p-ERK, ERK, PI3K, and GAPDH (Servicebio, Wuhan, China) at 4 °C overnight and were then incubated with secondary antibodies at 1:5000 dilution. Finally, signals were visualized by an enhanced ECL kit (Millipore, Burlington, MA, USA) (original western blot see Appendix A).

### 2.10. Cell Counting Kit (CCK)-8 Assay

Approximately 5.0 × 10^3^ cells per plate were seeded into 96-well plates with three replicates, and then incubated for 0, 24, 48, 72, and 96 h. 10% CCK-8 solution (10 μL) was added and incubated at 37 °C for 2 h. The absorbance values were measured at 450 nm by using microplate detector (SpectraMax iDS, Shanghai, China).

### 2.11. 5-Ethynyl-2-deoxyuridine (EdU) Assay

Briefly, 4 × 10^4^ cells were seeded into 24-well plates for one night and then incubated with 50 μM EdU buffer (RiboBio, Guangzhou, China) for 2 h. The cells were fixed with 4% formaldehyde for 10 min and permeabilized with 0.5% Triton X-100 for 15 min. Afterwards, the cells were stained with the Apollo reaction solution (200 μL) and Hoechst 33,342 (200 μL). The results were visualized by fluorescence microscopy (Olympus CKX53, Beijing, China).

### 2.12. Transwell Migration and Invasion Assay

4 × 10^4^ cells were plated into upper chamber (24-well insert; BD Biosciences, San Jose, CA, USA) with 200 μL of serum-free medium coated with or without Matrigel (1 mg/mL). The bottom chambers were filled with 500 μL RPMI 1640 containing 10% FBS. After 24 h of incubation, cells in the upper chamber were removed by scraping with a cotton swab. Invasive or migrating cells were fixed in 4% paraformaldehyde and stained with 0.1% crystal violet. The number of invasive or migrating cells was counted in five random fields under a microscope.

### 2.13. Statistical Analysis

Statistical tests were performed using SPSS 22.0 (IBM Corp., Armonk, NY, USA) and R software (version 4.0.2). Continuous vriables were expressed as the mean and range, while categorical variables were expressed as count and percentage. Univariate and multivariate Cox analysis were performed to identify independent prognostic factors and establish an integrated nomogram combining predictable clinicopathological factors and risk scores. The performance of model was assessed by the area under the receiver operating characteristic curve (AUC), concordance index (C-index), and a calibration curve. The clinical usefulness of the nomogram was evaluated using the decisions curve analysis (DCA). All tests were two-sided. The statistical significance was shown as follows: *p* < 0.05 (*), *p* < 0.01 (**), *p* < 0.001 (***).

## 3. Results

### 3.1. The Landscape of DR-lncRNA Regulators

To identify the functions and interaction of DR-lncRNA regulators in PTC, we analyzed the 58 normal tissues and 492 PTC tissues from the TCGA dataset and showed that most of the expressions of differentiation-related regulators were significantly lower in PTC, including *PAX8*, *SLC5A5*, *SLC5A8*, *SLC26A4*, *FOXE1*, *TG*, *TSHR*, *THRA*, *THRB*, *DIO1-2*, *GLIS3*, and *TPO* (all *p* < 0.001). Only *NKX2-1* significantly overexpressed in PTC tissues (Figure 2A). Next, the PPI network showed that TSHR was a hub gene which could interact with the other 15 genes (Appendix A). However, Pearson correlation analysis showed that *TSHR* had a weak correlation with other differentiation-related regulators. Interestingly, *DOUX1* and *DOUX2* had the strongest positive correlation (Appendix A). Furthermore, to study the relationship between differentiation-related regulators and lncRNA, Pearson correlation analysis was used to screen out DR-lncRNAs based on the criterion of Pearson’s coefficient |R| > 0.5 and *p* < 0.001. A total of 116 DR-lncRNAs were uncovered. Among them, five DR-lncRNAs were demonstrated to be associated with overall survival (*p* < 0.05). The LASSO analysis was used to further filter prognosis-related DR-lncRNAs (Appendix A). Finally, five prognostic related DR-lncRNAs, including *CASC15*, *LNC00900*, *AC055720-2*, *DPH6-DT*, and *TNRC6C-AS1*, still showed strong prognostic value. Figure 2B showed CASC15 and TNRC6C-AS1 were negatively related with differentiation-related regulators, whereas the other three lncRNAs showed positive correlation. Likewise, all DR-lncRNAs were abnormally expressed in PTC (*p* < 0.001, Figure 2C). The above results indicated that these lncRNAs may be a key regulator in the term of differentiation.

### 3.2. Risk Score Was Associated with Prognosis and Tumor Immune Microenvironment

To assess the prognostic value of DR-lncRNAs in PTC, a prognostic model was constructed. The risk score of each PTC patient was calculated according to the following formula: risk score = (2.43 × *CASC15*) + (−2.22 × *LNC00900*) + (−0.87 × *AC055720.2*) + (3.71 × *DPH6-DT*) + (0.28 × *TNRC6C-AS1*). All PTC patients were then divided into the high-risk and low-risk groups based on the median risk score. Patients in the low-risk group had a longer survival time than high-risk subgroups (*p* < 0.001, Figure 3A). The distribution of the OS, OS status, and risk score was displayed in Figure 3B. Heatmap distribution revealed that the expression levels of *LNC00900, AC055720.2,* and *TNRC6C-AS1* were significantly upregulated in the low-risk group, and a higher risk score was correlated with older age and higher tumor stage (Figure 3C). The AUC of risk score was 0.8742 (Figure 3D), indicating a good prediction performance. Regarding the immune microenvironment of PTC, CIBERSORT analysis revealed that the levels of eosinophils and activated DCs were significantly higher, whereas the levels of activated mast cells (MCs), resting MCs, and macrophages M2 were lower in the high-risk group compared with the low-risk group (Figure 4A). Utilizing the ssGSEA algorithm, we found T follicular helper cells, plasmacytoid DCs, immature DCs, and central memory CD8 T cells were plentiful in the low-risk group (Figure 4B). Additionally, MCP counter further confirmed the immune microenvironment’s association with risk score and revealed that the abundance of endothelial cells, neutrophils, and CD8+ T cells was distinctly higher in low-risk group (Figure 4C). These above data indicated risk score model could predict the prognosis and closely associated with the infiltration of immune cells.

In addition, the expression levels of PD-L1, TNFSF9, IDO2, and CD80 were significantly higher in the high-risk group than the low-risk group. However, patients with low-risk scores significantly elevated the expression level of PD-1, CD44, CD27, and CD160 (Appendix A). These discoveries suggested that risk scores might be used as a reference for different ICB therapies.

### 3.3. Functional and Pathway Enrichment Analysis

To further comprehend the biological mechanisms of DR-lncRNAs involved in PTC, we performed GO and KEGG pathway enrichment analyses with Metascape. The top nine GO terms were thyroid hormone generation, thyroid hormone metabolic process, hormone metabolic process, collagen-containing extracellular matrix, apical plasma membrane, basolateral plasma membrane, glycosaminoglycan binding, serine-type endopeptidase activity, and heparin binding (Appendix A–C). KEGG analysis showed that cell adhesion molecules (CAMs), ECM-receptor interaction, tryptophan metabolism, glycosaminoglycan biosynthesis, and keratan sulfate were closely involved in PTC development (Appendix A).

### 3.4. Drug Sensitivity Analysis

To determine the possible small molecules targeting differentiation-related regulators and further improve the clinical value of risk score model, we performed spearman correlation analysis to assess the correlation between drug sensitivity and differentiation-related regulators. Figure 5A showed the top 8 drugs with the most statistically significant differences (|Cor| ≥ 0.6 and *p* value < 0.01). We also compared the IC50 of drugs between the low- and high-risk groups. The results showed that the low-risk group was more sensitive to Bendamustine and TAS-6417 in targeting differentiation-related regulators than the high-risk group (Figure 5B). Appendix A shows the structure of the drugs. These findings highlight that the model could be considered as a potential chemosensitivity predictor.

### 3.5. Construction of the DR-lncRNAs Prognosis Nomogram

To quantitatively evaluate the prognosis of PTC patients in clinical practice, we constructed an integrated nomogram by combining the several predictable clinical factors with the DR-lncRNA-based risk scores. Univariate analysis showed that risk score (*p* < 0.001), M1 stage (*p* = 0.026), T4 stage (*p* = 0.003), TNM III—IV stage (*p* < 0.01), and age (*p* < 0.001) were significantly correlated with the overall survival (OS) in patients with PTC. Further multivariate analysis showed that risk score (OR = 2.38; 95% CI, 1.45–4.25; and *p* < 0.001) and age (OR = 1.15; 95% CI, 1.08–1.23; and *p* < 0.001) were the independent prognostic factors (Table 1). Subsequently, an integrated nomogram for the OS prediction was constructed (Figure 6A). The AUC of 3-, and 5-year OS was 0.966 and 0.967, respectively (Figure 6B). Additionally, the calibration plots revealed that the integrated nomogram model was excellent at predicting 3- and 5-year OS (Figure 6C). Moreover, DCA curves showed the integrated nomogram could better predict the OS and had a more favorable clinical applicability than either age or risk score (Figure 6D). Taken as a whole, these results revealed the significant value of the integrated nomogram in prognosticating patients with PTC.

### 3.6. Validation of the Expression of DR-lncRNAs

To validate the expression levels of DR-lncRNAs, we selected the GSE33630 database to conduct difference analysis between TC and normal tissues. The results showed that *LNC00900*, *AC055720-2*, and *TNRC6C-AS1* were upregulated in PTC tissues compared with ATC and normal tissues. Interestingly, the expression level of *DPH6-DT* significantly upregulated with an increase in differentiation degree. However, the expression level of *CASC15* was negative correlation with differentiation level (Appendix A), and patients with ATC have the highest risk scores (Appendix A). For subsequent molecular and functional experiments, we used TC tissues and cell lines to perform RT-qPCR analysis for further validation. As anticipated, the expression of *TNRC6C-AS1* and *CASC15* was significantly higher (*p* < 0.001), whereas *DPH6-DT* was significantly lower in TC tissues (Figure 7A) and cell lines (Figure 7B) than normal tissues and cell lines. Figure 7C shows that only *DPH6-DT* was associated with the level of differentiation. *TNRC6C-AS1* [39] and *CASC15* [40] have been studied in PTC. Therefore, we chose *DPH6-DT* for further experiments.

### 3.7. Downregulation of DPH6-DT Promote Proliferation and Metastasis by Activating the PI3K-AKT Signaling Pathway

To explore the biological functions of *DPH6-DT* in TC, *DPH6-DT* was knocked down via transfecting three siRNAs. Among them, *DPH6-DT*-siRNA3 efficiently knocked down *DPH6-DT* expression (Figure 8A). CCK-8 assay showed that silencing *DPH6-DT* conspicuously enhanced cell viability (Figure 8B), and EdU assay revealed that deficiency of *DPH6-DT* dramatically increased PTC cell proliferation in IHH-4 and KTC-1 cells (Figure 8C). Likewise, invasion and migration were enhanced when *DPH6-DT* silenced (Figure 8D). Furthermore, we explored the underlying mechanism changes of *DPH6-DT* -knockdown in PTC. Western blot analysis showed that the depletion of *DPH6-DT* could increase AKT, p-AKT, and PI3K expression levels in IHH-4 and KTC-1 cells. However, no difference was observed in ERK and p-ERK. Furthermore, *DPH6-DT* downregulation also increased the expression of vimentin and N-cadherin during epithelial-mesenchymal transition (EMT). These data suggested that ablation of *DPH6-DT* led to the activation of PI3K/AKT signaling pathway in PTC.

## 4. Discussion

Over the past decade, RAI therapy has been considered an effective treatment that successfully ablates any metastatic tumor and residual thyroid tissue and contributes to excellent prognosis in patients with PTC [41]. However, dedifferentiation is the main concern associated with RAI refractoriness and dramatically decreased RAI uptake, which increase the risk of recurrence and PTC-related mortality [42]. Unfortunately, there are no definitive strategies to restore RAI avidity through thyroid-specific genes. Thus, our study utilized bioinformatic analysis to identify the prognostic significance of DR-lncRNAs, including *CASC15*, *LNC00900*, *AC055720-2*, *DPH6-DT*, and *TNRC6C-AS1*, which are closely associated with tumor differentiation. The corresponding risk score was established based on these DR-lncRNAs acting as a potential prognostic and chemosensitivity predictor. Moreover, the in vitro experiments further validate downregulation of *DPH6-DT* and promote proliferation and metastasis by activating the PI3K-AKT signaling pathway (Figure 9). These findings indicate that the signature can not only serve as a useful indicator of dedifferentiation, but also predict drug sensitivity and poor clinical outcomes in patients with PTC.

In recent years, numerous studies [43,44] have confirmed that tumor immune microenvironment plays a decisive role in tumor progression and therapeutic efficacy through the interaction and coevolution of the tumor stroma, immune cells, and tumor cells. Previous mainstream studies [23,45] have demonstrated that Tregs, TAMs, MCs, DCs, and neutrophils exert a tumor-promoting effect, and NK cells, CD8^+^ T cells, and B cells play an anti-tumor role. Our research used CIBERSORT, ssGSEA, and MCP counter to evaluate the level of immune cell infiltration to verify the previously reported results of immune cell functions and phenotypes in patients with PTC from the perspective of proportion and abundance (Figure 4), and partially supports the evidence that T follicular helper cell, central memory CD8 T cell, and immature DC were significantly increased in the low-risk group to exert an anti-tumor effect. Furthermore, immune checkpoints were considered as promising targets for PTC treatment that had entered clinical trials, including inhibition of PD-L1 (avelumab, atezolizumab, durvalumab) and anti-PD-1 (pembrolizumab, nivolumab) [46]. An earlier study by Chowdhury [47] confirmed that the expression of PD-L1 was associated with aggressive clinicopathological markers and significantly reduced disease-free survival in PTC. Anti-PD-L1 therapy (pembrolizumab) has generated potential clinical benefits for advanced differentiated TC patients by inducing regression of aggressive tumors. Our research revealed that PD-L1, TNFSF9, IDO2, CD80, and CD70 were significantly higher in the high-risk group than the low-risk group. These discoveries suggested that risk scores might be used as a reference for different ICB therapies.

In addition, conventional anti-cancer treatments (radio- and chemotherapy) have been explored with the purpose of restoring RAI sensitivity in poorly differentiated TC patients. Initially, the PPARγ activators [48], histone deacetylase inhibitors [49], and inhibitors of iodide release [50] were demonstrated to be minimally effective in clinical trials. Recently, RAI treatment combined with inhibitors of MEK and BRAF kinases significantly improved clinical responses for RAI-refractory TC patients [27]. However, not all poorly differentiated TC patients benefit from the adjunctive treatments that meet the expectation for restoring RAI sensitivity. Therefore, additional efforts need to further optimize the clinical efficacy to induce TC redifferentiation. Functional and pathway enrichment analysis were performed to further comprehend the biological mechanisms about DR-lncRNA and revealed that CAMs, ECM-receptor interaction, tryptophan metabolism, glycosaminoglycan biosynthesis, and keratan sulfate were closely involved in PTC development. Moreover, the CellMiner database was widely used for cancer drug testing, which involved 20,503 compounds and 22,379 genes [38], and provides a new perspective of the treatment for TC patients with RAI-refractory TC. Our results demonstrated that Bendamustine and TAS-6417 were more sensitive in targeting differentiation-related regulators in the low-risk group than in the high-risk group. Among these agents, Bendamustine alone or combined with other antineoplastic agents was widely used in treating chronic lymphocytic leukemia and refractory Hodgkin lymphoma [51]. Bendamustine and TAS-6417 have not been approved for TC treatment and their clinical efficacy is unknown to date. Therefore, large-cohort prospective clinical trials are necessary to validate drug efficacy in future research.

In the further analyses, we constructed an integrated nomogram by combining DR-lncRNAs risk scores with age to improve the accuracy of the prognostic prediction and risk stratification. The AUC and calibration plots indicated the nomogram model has advantageous usability in survival prediction. Currently, the TNM 8th edition was the most widely used to assess the prognosis of PTC and achieve great advancement in clinical practice [52], but it is difficult to distinguish patients with similar clinicopathologic characteristics for different survival outcomes, which caused low-risk patients to adopt a higher degree of TSH inhibition and unnecessary RAI treatment. Our study has discovered some prognosis-related DR-lncRNAs which contributed to prognostic judgement and decision-making for clinical treatment.

Finally, we validated the role of DR-lncRNAs in TC cell growth, proliferation, and dedifferentiation. Several studies [42,53] have reported some molecules and pathways involved in the dedifferentiation process of TC, including *MAPK, P53, BRAF, EIF1AX*, and histone methyltransferases. Ma and colleagues [54] demonstrated that the metabolic gene signature can be used as an indicator of the dedifferentiation biomarker for PTC. Similarly, Suh et al. [55] found that the deregulations of glucose metabolism could mediate dedifferentiation of PTC. Sara C. et al. [56] highlighted the *FOXE1* as a novel differentiation-related gene which induces epithelial-to-mesenchymal and cell proliferation. In addition, genetic alterations in *PI3K/AKT* and *MAPK* signaling pathways by chromosomal rearrangements or point mutations are vital drivers to silence the expression of differentiation-related genes, resulting in loss of RAI avidity [41,57]. Our study showed that *DPH6-DT* was significantly lower in TC tissues than normal tissues, and most of the expression of differentiation-related regulators were also significantly lower in TC. Figure 1 uncovered that the expression level of *DHP6-DT* was positively associated with differentiation-related regulators. At the same time, we validated that the expression level of *DPH6-DT* was significantly linked to TC differentiation degree (Figure 7). Silencing *DPH6-DT* dramatically promoted cell proliferation and metastasis by activating the *PI3K-AKT* signaling pathway. The *PI3K-AKT* pathway has been well recognized as regulating cell differentiation via decreasing the expression of NIS in PTC. To our knowledge, this is the first study that revealed *DPH6-DT* could act as a useful signature to indicate differentiation and uncover the underlying molecular mechanism.

Undeniably, there are several limitations in the current study. First, the expression level of DR-lncRNAs was only validated in PTC and normal tissues, and it is difficult to acquire anaplastic and RAI-refractory TC tissue to verify the differentiation grade due to its low prevalence. Second, the integrated nomogram showed good performance in predicting prognosis. However, there are no additional databases to use for external validation. Future clinical research is necessary to confirm the validity of our survival prediction model. Third, because of the limited project funding, we performed functional and molecular experiments to uncover the underlying mechanism of PTC differentiation in vitro. The role of *DPH6-DT* and related pathways should be studied in depth in the context of differentiation in in vivo experiments. Lastly, the TCGA database mainly records the RNA-Seq and clinical data of European and North American populations [58], resulting in an inevitable selection bias.

In summary, this study systematically delineated the expression pattern, tumor immune microenvironment, drugs sensitivity, and prognostic value of *DR-*lncRNAs regulators, and revealed its great significance in prognosis prediction and clinical treatment strategy in patients with PTC. More importantly, we confirm that *DPH6-DT* could be considered as a novel signature to indicate differentiation and promote TC progression via activating the *PI3K-AKT* signaling pathway, which provides crucial insight in early diagnosis and therapy to improve the poor clinical outcome of anaplastic and RAI-refractory TC patients.

## Figures and Tables

**Figure 1 cancers-14-01353-f001:**
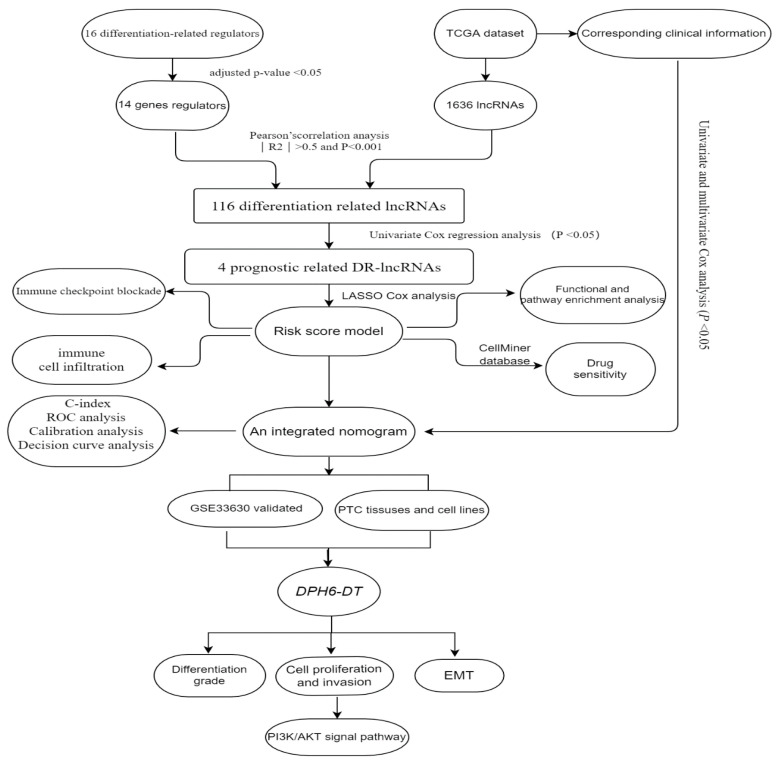
Study flow chart.

**Figure 2 cancers-14-01353-f002:**
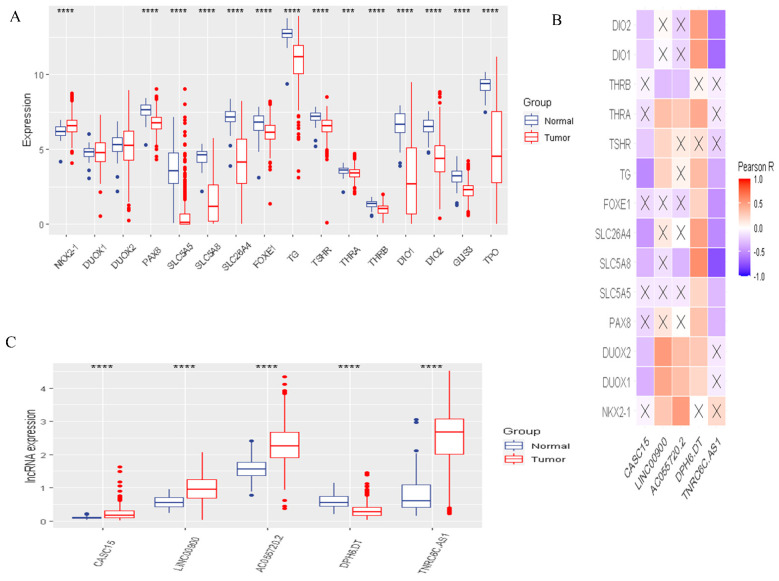
The landscape of DR-lncRNAs regulators. (**A**) Expression of 16 differentiation related regulators in normal and tumor samples; (**B**) The relationship between differentiation-related regulators and DR-lncRNAs. (**C**) Differential expression of 5 DR-lncRNAs regulators. *** *p* < 0.001, **** *p* < 0.0001.

**Figure 3 cancers-14-01353-f003:**
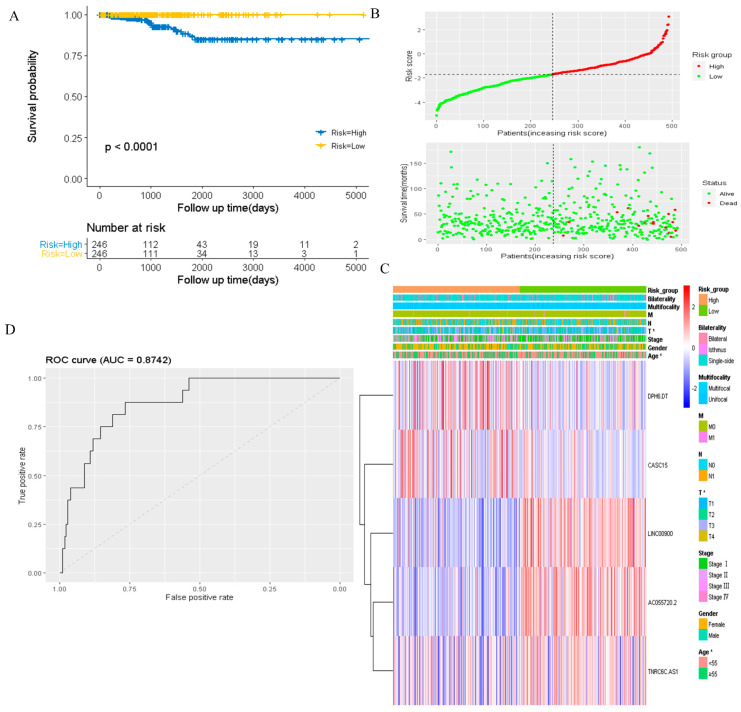
Prognostic value of risk score. (**A**) Survival analysis of patients in the low-risk and high-risk groups. (**B**) Distributions of survival status and risk scores. (**C**) Heatmap distribution of risk scores and clinicopathological characteristics of the two groups. (**D**) The AUC of the risk score. AUC: the area under the receiver operating characteristic curve.

**Figure 4 cancers-14-01353-f004:**
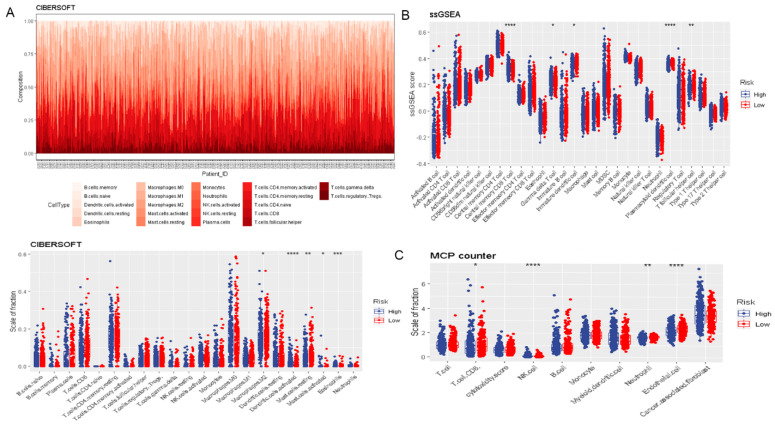
The correlation between risk signature and immune cell infiltration. (**A**) CIBERSORT. (**B**) ssGSEA. (**C**) MCP counter. **** *p* < 0.0001, *** *p* < 0.001, ** *p* < 0.01, and * *p* < 0.05.

**Figure 5 cancers-14-01353-f005:**
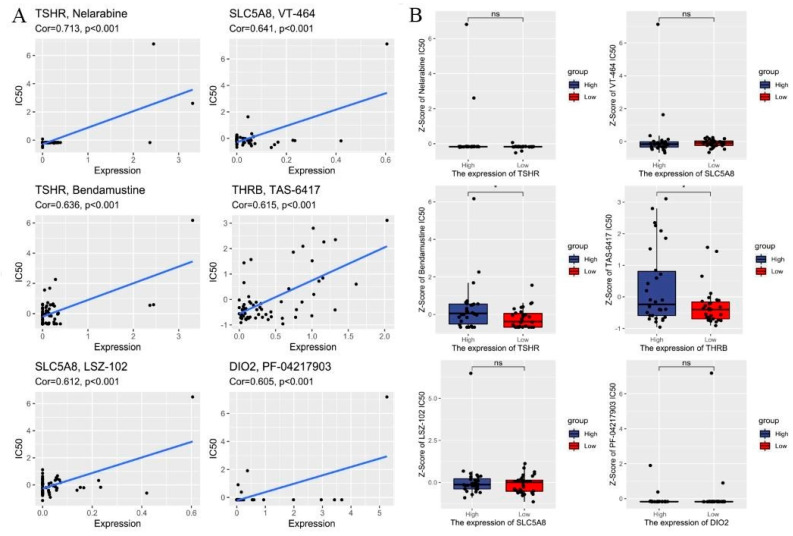
Drug sensitivity analysis. (**A**) Correlations between IC50 for different drugs and differentiation-related regulators. (**B**) Compared the efficiency of the chosen drugs in relation to the risk score. * *p* < 0.05 and ns: no significance.

**Figure 6 cancers-14-01353-f006:**
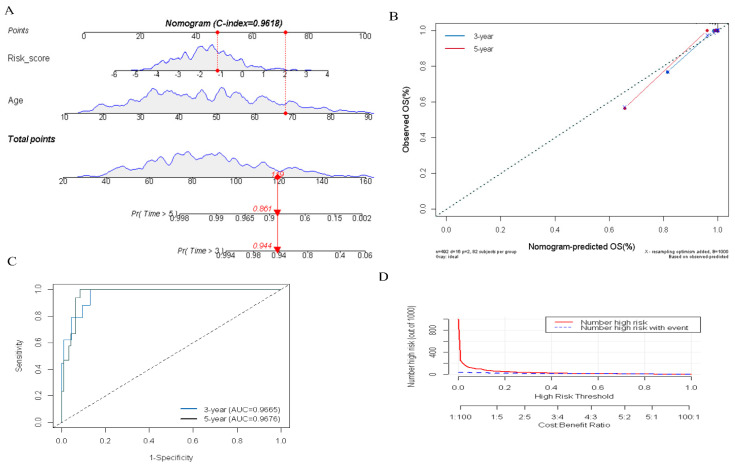
Construction and evaluation of an integrated nomogram. (**A**) Nomogram was developed based on DR-lncRNA-based risk scores and age. (**B**) Calibration plots were performed to evaluate the predictive performance of 3- and 5-year OS. (**C**) The AUC value of the nomogram. (**D**) DCA of the nomogram. OS: overall survival, AUC: the area under the receiver operating characteristic curve. DCA: decisions curve analysis.

**Figure 7 cancers-14-01353-f007:**
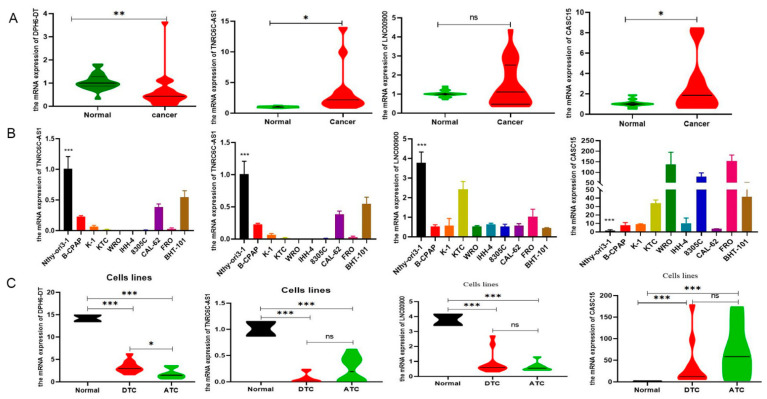
Differential expression of prognostic related DR-lncRNAs. (**A**) Twenty paired PTC samples and adjacent normal tissues. (**B**) Nine thyroid cells and normal thyroid follicular epithelial cells. (**C**) PTC, ATC, and normal thyroid follicular epithelial cells. ATC: anaplastic thyroid cancer. PTC: papillary thyroid cancer. *** *p* < 0.001, ** *p* < 0.01, * *p* < 0.05 and no significance.

**Figure 8 cancers-14-01353-f008:**
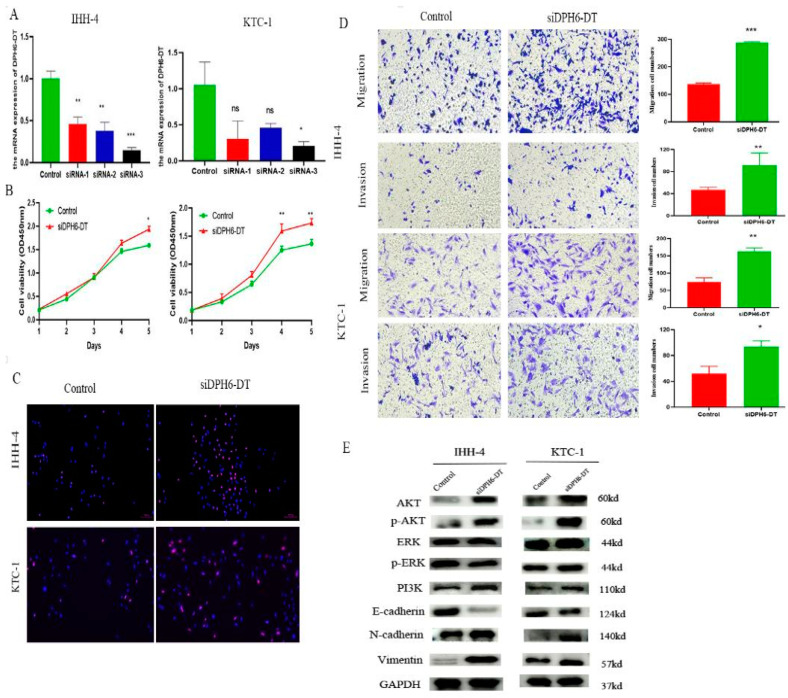
Depletion of *DHP6-DT* promoted proliferation and metastasis by activating the PI3K-AKT signaling pathway. (**A**) IHH-4 and KTC-1 cells were transfected with different si*DPH6-DT* and scramble vector (control). (**B**) Knockdown of *DPH6-DT* enhanced cell viability by CCK-8 assay. (**C**) Deficiency of *DPH6-DT* dramatically increased cell proliferation by EdU assay. (**D**) Knockdown of *DPH6-DT* accelerated cell migration and invasion. (**E**) Western blot for the EMT and PI3K-AKT signal pathway related protein expression upon the knockdown of DPH6-DT. EMT: epithelial-mesenchymal transition. *** *p* < 0.001, ** *p* < 0.01, and * *p* < 0.05.

**Figure 9 cancers-14-01353-f009:**
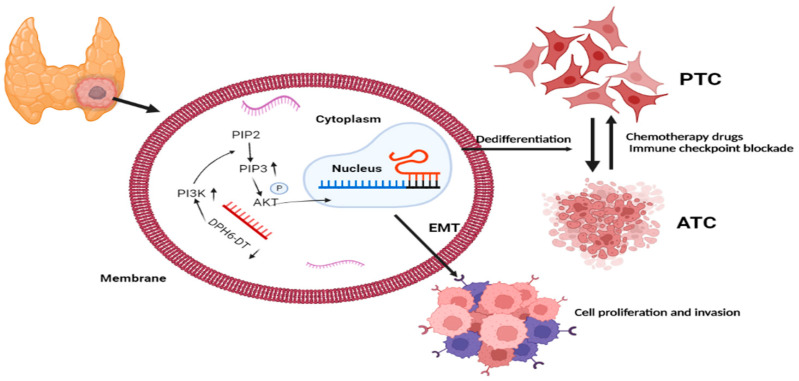
Brief summary for the molecular mechanism of DR-lncRNAs. EMT: epithelial-mesenchymal transition. ATC: anaplastic thyroid cancer. PTC: papillary thyroid cancer. Biorender, available online: https://app.biorender.com (accessed on 2 November 2021).

**Table 1 cancers-14-01353-t001:** Univariate and multivariate Cox regression analysis the prognosis factors of PTC.

Variable Total (%) N = 492	Univariate Analysis		Multivariate Analysis	
	HR (95%CI)	*p*	OR (95%CI)	*p*
Age 47.01 ± 16.03	1.16 (1.10–1.22)	<0.001	1.15 (1.08–1.23)	<0.001
Gender Female 361 (77.37) Male 131 (26.63)	Ref. 1.92 (0.69–5.29)	0.21	-	-
TNM Stage I 276 (56.10) II 52 (10.57) III 109 (22.15) IV 55 (11.18)	Ref. 5.67 (0.80–40.27) 10.25 (2.13–49.45) 16.47 (3.18–85.33)	0.083 0.004 <0.001	Ref. 0.77 (0.01–63.96) 0.62 (0.01–47.80) 0.10 (0–15.71)	0.906 0.827 0.375
T Stage T1 135 (27.44) T2 162 (32.93) T3 172 (34.96) T4 23 (4.67)	Ref. 1.10 (0.18–6.59) 1.69 (0.33–8.73) 11.52 (2.31–57.57)	0.920 0.531 0.003	Ref. 1.49 (0.02–100.44) 0.59 (0.01–41.61) 13.61 (0.11–1620.48)	0.852 0.809 0.284
N Stage N0 271 (55.08) N1 221 (44.92)	Ref. 1.14 (0.43–3.05)	0.078	-	-
M Stage M0 483 (98.17) M1 9 (1.83)	Ref. 5.39 (1.22–23.83)	0.026	Ref. 0.51 (0.05–4.68)	0.55
Multifocality Multifocal 228 (46.34) Unifocal 264 (53.66)	Ref. 3.91 (0.88–17.34)	0.073	-	-
Bilaterality Bilateral 84 (17.07) Unilateral 386 (78.46) Isthmus 22 (4.47)	Ref. 0.95 (0.21–4.26) 1.05 (0.09–11.79)	0.942 0.967	-	-
Risk score	2.99 (2.08–4.31)	<0.001	2.38 (1.45–4.25)	<0.001

Abbreviations: CI: confidence intervals, HR: hazard ratio, OR: odds ratio.

## Data Availability

The original contributions presented in the study are included in the article/Appendix A. Further inquiries can be directed to the corresponding authors.

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
