# Peer review of "Comprehensive Analysis of the Prognosis and Drug Sensitivity of Differentiation-Related lncRNAs in Papillary Thyroid Cancer"

_cancers, 2022, doi:10.3390/cancers14051353_

Round 1

Reviewer 1 Report

The title is interesting and it is going to examine role of lncRNAs in thyroid cancer. This experiment evaluates in vitro and bioinformatics analysis to confirm their results. The title is novel and it examines drug sensitivity and prognosis of thyroid cancer. The lncRNA is a hot topic. I suggest publication of current manuscript. The references should be updated and new references should be added to improve visibility and quality of manuscript. The first paragraph is about thyroid cancer, but it provides general information. Authors should provide more information about malignancy and chemo-resistance in thyroid cancer based on new publications. The second paragraph of introduction should be separated and divided into two sections to prevent confusion of readers. The part related to lncRNAs is not enough and should be improved about localization, association with other pathways such as miRNAs, role in cancer progression and drug resistance. Suggested articles could be Doi, 10.1016/j.canlet.2021.03.025 and Doi, 10.1016/j.canlet.2021.03.018. Divide discussion to different sections section to make it easier for readers to understand contents.

Author Response

We would like to thank you for your thoughtful comments on our manuscript. We believe that the additional changes we have made in response to your comments have improved the comprehension of the study put forth in the manuscript. Below is our point-by-point response to your comments. We have not listed all minor changes made below. However, revised parts are highlighted in red in the manuscript.

Comment 1: The first paragraph is about thyroid cancer, but it provides general information. Authors should provide more information about malignancy and chemo-resistance in thyroid cancer based on new publications.

Response: Thank you for your constructive suggestions. The malignancy and chem-resistance in thyroid cancer were elaborated detailed in the section of discussion in line 376-395. Thank you again. 

Comment 2: The second paragraph of introduction should be separated and divided into two sections to prevent confusion of readers. The part related to lncRNAs is not enough and should be improved about localization, association with other pathways such as miRNAs, role in cancer progression and drug resistance. Suggested articles could be Doi, 10.1016/j.canlet.2021.03.025 and Doi, 10.1016/j.canlet.2021.03.018. Divide discussion to different sections section to make it easier for readers to understand contents.

Response: We feel great thanks for your valuable and thoughtful comments, for this part of content, we conducted an in-depth analysis as follows:

Recently, an increasing number of studies have reported that genetic and epigenetic aberrations[1, 2], cancer stem cells[3], microRNAs[4], immunometabolic networks[5], and autophagy[6,7] play a critical role in PTC dedifferentiation and RAI resistance. Long non-coding RNAs (lncRNA), defined as a series of transcripts greater than 200 nucleotides, have been regarded as crucial regulators at various levels of transcriptional, post-transcriptional, and translational regulation. Extensively involved in carcinogenesis, chromatin dynamics, interactions with mRNAs and protein, differentiation and embryonic development in patients with TC[8, 9]. Noteworthy, several studies proved that lncRNA could effectively modulate NF-κB and PI3K-AKT signaling pathway thus affecting tumorigenesis[10, 11], and lncRNACDC6 severed as ceRNA to target CDC6 via sponging micro-225 to promote breast cancer progression[12]. Meantime, impairment or disturbances in lncRNA expression resulting in increased chemoresistance[13]. lncRNA CRNDE directly binding to SRSF6 and reducing the alternative splicing of PICALM, thereby mediating chemoresistance in gastric cancer[14]. Therefore, lncRNA can be considered as a prognostic and therapeutic marker for cancer.

Besides, some studies [15-18] revealed that lncRNA served as potentially useful biomarker for the malignant thyroid nodule diagnoses, prognosis prediction, and treatment response of PTCs. Our previous study demonstrated that PTCSC3, as a tumor suppressor, was associated with prognosis in TC[17]. A systematic review and meta-analysis showed that lncRNA function as biomarker for TC diagnosis and lymph nodes metastasis prediction[19]. However, differentiation-related lncRNA (DR-lncRNA) mediating PTC dedifferentiation patterns are not yet fully elucidated.

We tried our best to improve the manuscript and made some changes in the manuscript.  These changes will not influence the content and framework of the paper. And marked in red in revised paper.

We appreciate for your warm work earnestly, and hope that the correction will meet with approval.

Once again, thank you very much for your comments and suggestions.

  1. Cheng W;Liu R;Zhu G;Wang H, Xing M: Robust Thyroid Gene Expression and Radioiodine Uptake Induced by Simultaneous Suppression of BRAF V600E and Histone Deacetylase in Thyroid Cancer Cells. J Clin Endocrinol Metab.2016; 101:962-971.
  2. Hardin H;Montemayor-Garcia C, Lloyd RV: Thyroid cancer stem-like cells and epithelial-mesenchymal transition in thyroid cancers. Hum Pathol.2013; 44:1707-1713.
  3. Riesco-Eizaguirre G;Wert-Lamas L;Perales-Paton J;Sastre-Perona A;Fernandez LP, Santisteban P: The miR-146b-3p/PAX8/NIS Regulatory Circuit Modulates the Differentiation Phenotype and Function of Thyroid Cells during Carcinogenesis. Cancer Res.2015; 75:4119-4130.
  4. Galdiero MR;Varricchi G, Marone G: The immune network in thyroid cancer. Oncoimmunology.2016; 5:e1168556.
  5. Gewirtz DA: An autophagic switch in the response of tumor cells to radiation and chemotherapy. Biochem Pharmacol.2014; 90:208-211.
  6. Jin SM;Jang HW;Sohn SY;Kim NK;Joung JY;Cho YY;Kim SW, Chung JH: Role of autophagy in the resistance to tumour necrosis factor-related apoptosis-inducing ligand-induced apoptosis in papillary and anaplastic thyroid cancer cells. Endocrine.2014; 45:256-262.
  7. Schaukowitch K, Kim TK: Emerging epigenetic mechanisms of long non-coding RNAs. Neuroscience.2014; 264:25-38.
  8. Sun M, Kraus WL: From discovery to function: the expanding roles of long noncoding RNAs in physiology and disease. Endocr Rev.2015; 36:25-64.
  9. Mirzaei S;Zarrabi A;Hashemi F;Zabolian A;Saleki H;Ranjbar A;Seyed Saleh SH;Bagherian M;Sharifzadeh SO;Hushmandi Ket al: Regulation of Nuclear Factor-KappaB (NF-kappaB) signaling pathway by non-coding RNAs in cancer: Inhibiting or promoting carcinogenesis? Cancer Lett.2021; 509:63-80.
  10. Zhang W;Yang S;Chen D;Yuwen D;Zhang J;Wei X;Han X, Guan X: SOX2-OT induced by PAI-1 promotes triple-negative breast cancer cells metastasis by sponging miR-942-5p and activating PI3K/Akt signaling. Cell Mol Life Sci.2022; 79:59.
  11. Kong X;Duan Y;Sang Y;Li Y;Zhang H;Liang Y;Liu Y;Zhang N, Yang Q: LncRNA-CDC6 promotes breast cancer progression and function as ceRNA to target CDC6 by sponging microRNA-215. J Cell Physiol.2019; 234:9105-9117.
  12. Ashrafizaveh S;Ashrafizadeh M;Zarrabi A;Husmandi K;Zabolian A;Shahinozzaman M;Aref AR;Hamblin MR;Nabavi N;Crea Fet al: Long non-coding RNAs in the doxorubicin resistance of cancer cells. Cancer Lett.2021; 508:104-114.
  13. Zhang F;Wang H;Yu J;Yao X;Yang S;Li W;Xu L, Zhao L: LncRNA CRNDE attenuates chemoresistance in gastric cancer via SRSF6-regulated alternative splicing of PICALM. Mol Cancer.2021; 20:6.
  14. Zhou H;Sun Z;Li S;Wang X, Zhou X: LncRNA SPRY4-IT was concerned with the poor prognosis and contributed to the progression of thyroid cancer. Cancer Gene Ther.2018; 25:39-46.
  15. Li HM;Yang H;Wen DY;Luo YH;Liang CY;Pan DH;Ma W;Chen G;He Y, Chen JQ: Overexpression of LncRNA HOTAIR is Associated with Poor Prognosis in Thyroid Carcinoma: A Study Based on TCGA and GEO Data. Horm Metab Res.2017; 49:388-399.
  16. Fan M;Li X;Jiang W;Huang Y;Li J, Wang Z: A long non-coding RNA, PTCSC3, as a tumor suppressor and a target of miRNAs in thyroid cancer cells. Exp Ther Med.2013; 5:1143-1146.
  17. Mahmoudian-Sani MR;Jalali A;Jamshidi M;Moridi H;Alghasi A;Shojaeian A, Mobini GR: Long Non-Coding RNAs in Thyroid Cancer: Implications for Pathogenesis, Diagnosis, and Therapy. Oncol Res Treat.2019; 42:136-142.
  18. Jing W;Li X;Peng R;Lv S;Zhang Y;Cao Z;Tu J, Ming L: The diagnostic and prognostic significance of long noncoding RNAs expression in thyroid cancer: A systematic review and meta-analysis. Pathol Res Pract.2018; 214:327-334.

Reviewer 2 Report

The current manuscript focuses on lncRNAs in thyroid cancer. The cancer is the second leading cause of death after cardiovascular diseases and understanding underlying mechanisms involved in its progression are of importance. Furthermore, lncRNAs are hot topic nowadays and due to their capacity in regulating various biological events, studies have focused on revealing their function in cancer. I suggest publication of current manuscript. However, some issues as following should be addressed. Please use template of cancers journal. Please add "non-coding RNA" as a keyword. Add more references from 2020 and 2021 to improve quality and visibility of current manuscript. if it is possible, please add a schematic figure about contents of manuscript to make it more attractive for readers. Add more specific information about thyroid cancer in first paragraph of introduction and make it more clear that molecular pathways are involved in its progression. Make space between words and references at the end of each statement or line. The part related to lncRNAs in introduction is too general and adds nothing to field. Discuss more about critical functions in lncRNAs in cancer and how they can interact with other molecular pathways. Suggested article (Doi, 10.1016/j.canlet.2021.03.025). Check all sections of in terms of spell and grammar mistakes.

Author Response

We would like to thank you for your thoughtful comments on our manuscript. We believe that the additional changes we have made in response to your comments have improved the comprehension of the study put forth in the manuscript. Below is our point-by-point response to your comments. We have not listed all minor changes made below. However, revised parts are highlighted in red in the manuscript.

Comment 1: Please use template of cancers journal. Please add "non-coding RNA" as a keyword. Add more references from 2020 and 2021 to improve quality and visibility of current manuscript.

Response: Thank you for your constructive suggestions. According to your nice suggestion, we add "long non-coding RNA" as a keyword and increase some references from 2020 and 2021 to improve quality and visibility of current manuscript. Thank your very much.

Comment 2: if it is possible, please add a schematic figure about contents of manuscript to make it more attractive for readers.

Response: we greatly admire your nice advice. In our study, the figure 1 represent the study flow chart, which make readers easily understand the idea and structure of the article. Meantime, the figure 9 shows brief summary for the molecular mechanism of DR-lncRNAs, which  make it more attractive for readers.

Comment 3: Add more specific information about thyroid cancer in first paragraph of introduction and make it more clear that molecular pathways are involved in its progression.

Response: We feel great thanks for your valuable and thoughtful comments. For this part of content, The malignancy and chem-resistance in thyroid cancer were elaborated detailed in the section of discussion in line 376-395.

 Comment 4:The part related to lncRNAs in introduction is too general and adds nothing to field. Discuss more about critical functions in lncRNAs in cancer and how they can interact with other molecular pathways. Suggested article (Doi, 10.1016/j.canlet.2021.03.025). Check all sections of in terms of spell and grammar mistakes.

Response: According to your valuable and thoughtful comments, for this part of content, we conducted an in-depth analysis as follows:

Recently, an increasing number of studies have reported that genetic and epigenetic aberrations[1, 2], cancer stem cells[3], microRNAs[4], immunometabolic networks[5], and autophagy[6,7] play a critical role in PTC dedifferentiation and RAI resistance. Long non-coding RNAs (lncRNA), defined as a series of transcripts greater than 200 nucleotides, have been regarded as crucial regulators at various levels of transcriptional, post-transcriptional, and translational regulation. Extensively involved in carcinogenesis, chromatin dynamics, interactions with mRNAs and protein, differentiation and embryonic development in patients with TC[8, 9]. Noteworthy, several studies proved that lncRNA could effectively modulate NF-κB and PI3K-AKT signaling pathway thus affecting tumorigenesis[10, 11], and lncRNACDC6 severed as ceRNA to target CDC6 via sponging micro-225 to promote breast cancer progression[12]. Meantime, impairment or disturbances in lncRNA expression resulting in increased chemo-resistance[13]. lncRNA CRNDE directly binding to SRSF6 and reducing the alternative splicing of PICALM, thereby mediating chemoresistance in gastric cancer[14]. Therefore, lncRNA can be considered as a prognostic and therapeutic marker for cancer.

Besides, some studies [15-18] revealed that lncRNA served as potentially useful biomarker for the malignant thyroid nodule diagnoses, prognosis prediction, and treatment response of PTCs. Our previous study demonstrated that PTCSC3, as a tumor suppressor, was associated with prognosis in TC[17]. A systematic review and meta-analysis showed that lncRNA function as biomarker for TC diagnosis and lymph nodes metastasis prediction[19]. However, differentiation-related lncRNA (DR-lncRNA) mediating PTC dedifferentiation patterns are not yet fully elucidated.

We tried our best to improve the manuscript and made some changes in the manuscript.These changes will not influence the content and framework of the paper. And marked in red in revised paper. We invited a language company (MedEiditing LLC) to check all sections of in terms of spell and grammar mistakes. We appreciate for your warm work earnestly, and hope that the correction will meet with approval.

Once again, thank you very much for your comments and suggestions.

  1. Cheng W;Liu R;Zhu G;Wang H, Xing M: Robust Thyroid Gene Expression and Radioiodine Uptake Induced by Simultaneous Suppression of BRAF V600E and Histone Deacetylase in Thyroid Cancer Cells. J Clin Endocrinol Metab.2016; 101:962-971.
  2. Hardin H;Montemayor-Garcia C, Lloyd RV: Thyroid cancer stem-like cells and epithelial-mesenchymal transition in thyroid cancers. Hum Pathol.2013; 44:1707-1713.
  3. Riesco-Eizaguirre G;Wert-Lamas L;Perales-Paton J;Sastre-Perona A;Fernandez LP, Santisteban P: The miR-146b-3p/PAX8/NIS Regulatory Circuit Modulates the Differentiation Phenotype and Function of Thyroid Cells during Carcinogenesis. Cancer Res.2015; 75:4119-4130.
  4. Galdiero MR;Varricchi G, Marone G: The immune network in thyroid cancer. Oncoimmunology.2016; 5:e1168556.
  5. Gewirtz DA: An autophagic switch in the response of tumor cells to radiation and chemotherapy. Biochem Pharmacol.2014; 90:208-211.
  6. Jin SM;Jang HW;Sohn SY;Kim NK;Joung JY;Cho YY;Kim SW, Chung JH: Role of autophagy in the resistance to tumour necrosis factor-related apoptosis-inducing ligand-induced apoptosis in papillary and anaplastic thyroid cancer cells. Endocrine.2014; 45:256-262.
  7. Schaukowitch K, Kim TK: Emerging epigenetic mechanisms of long non-coding RNAs. Neuroscience.2014; 264:25-38.
  8. Sun M, Kraus WL: From discovery to function: the expanding roles of long noncoding RNAs in physiology and disease. Endocr Rev.2015; 36:25-64.
  9. Mirzaei S;Zarrabi A;Hashemi F;Zabolian A;Saleki H;Ranjbar A;Seyed Saleh SH;Bagherian M;Sharifzadeh SO;Hushmandi Ket al: Regulation of Nuclear Factor-KappaB (NF-kappaB) signaling pathway by non-coding RNAs in cancer: Inhibiting or promoting carcinogenesis? Cancer Lett.2021; 509:63-80.
  10. Zhang W;Yang S;Chen D;Yuwen D;Zhang J;Wei X;Han X, Guan X: SOX2-OT induced by PAI-1 promotes triple-negative breast cancer cells metastasis by sponging miR-942-5p and activating PI3K/Akt signaling. Cell Mol Life Sci.2022; 79:59.
  11. Kong X;Duan Y;Sang Y;Li Y;Zhang H;Liang Y;Liu Y;Zhang N, Yang Q: LncRNA-CDC6 promotes breast cancer progression and function as ceRNA to target CDC6 by sponging microRNA-215. J Cell Physiol.2019; 234:9105-9117.
  12. Ashrafizaveh S;Ashrafizadeh M;Zarrabi A;Husmandi K;Zabolian A;Shahinozzaman M;Aref AR;Hamblin MR;Nabavi N;Crea Fet al: Long non-coding RNAs in the doxorubicin resistance of cancer cells. Cancer Lett.2021; 508:104-114.
  13. Zhang F;Wang H;Yu J;Yao X;Yang S;Li W;Xu L, Zhao L: LncRNA CRNDE attenuates chemoresistance in gastric cancer via SRSF6-regulated alternative splicing of PICALM. Mol Cancer.2021; 20:6.
  14. Zhou H;Sun Z;Li S;Wang X, Zhou X: LncRNA SPRY4-IT was concerned with the poor prognosis and contributed to the progression of thyroid cancer. Cancer Gene Ther.2018; 25:39-46.
  15. Li HM;Yang H;Wen DY;Luo YH;Liang CY;Pan DH;Ma W;Chen G;He Y, Chen JQ: Overexpression of LncRNA HOTAIR is Associated with Poor Prognosis in Thyroid Carcinoma: A Study Based on TCGA and GEO Data. Horm Metab Res.2017; 49:388-399.
  16. Fan M;Li X;Jiang W;Huang Y;Li J, Wang Z: A long non-coding RNA, PTCSC3, as a tumor suppressor and a target of miRNAs in thyroid cancer cells. Exp Ther Med.2013; 5:1143-1146.
  17. Mahmoudian-Sani MR;Jalali A;Jamshidi M;Moridi H;Alghasi A;Shojaeian A, Mobini GR: Long Non-Coding RNAs in Thyroid Cancer: Implications for Pathogenesis, Diagnosis, and Therapy. Oncol Res Treat.2019; 42:136-142.
  18. Jing W;Li X;Peng R;Lv S;Zhang Y;Cao Z;Tu J, Ming L: The diagnostic and prognostic significance of long noncoding RNAs expression in thyroid cancer: A systematic review and meta-analysis. Pathol Res Pract.2018; 214:327-334.

Reviewer 3 Report

In the present paper the authors investigated the prognostic role of differentiation-related (DR) lncRNAs in papillary thyroid cancer (PTC). Five prognostic related DR lncRNAs were selected to establish a prognostic-predicting model and corresponding risk scores was associated with overall survival, immune infiltration and drug sensitivity. The data obtained are promising, suggesting a role of the identified lncRNAs signature to evaluate prognosis and best therapeutic options for patients with PTC. However, several critical points are present:

The selection of the initial 1636 lncRNAs should be better explained.

Line 167-170 and Figures 2B and 2C: These results are not related with the topic and may be moved in the supplemental data.

Figure 2C and 2D: Explain the meaning of the crosses.

In contrast to the sentence at lines 173-175, Figure 2D showed a negative correlation of CASC15 and TNRC6C-AS1 with differentiation-related regulators and a positive correlation for the other three lncRNAs. Moreover, all lncRNAs except DPH6-DT were increased in PTCs, therefore I would expect their negative correlation with differentiation-related regulators.

Paragraph 3.2: The five prognosis-related DR-lncRNAs were already cited in paragraph 3.1. In Fig. S1 it is not clear their significant correlation with prognosis. What does represent the upper panel of Figure 3B (ratio score/increasing risk score)?

DPH6-DT expression was positively correlated with thyroid differentiation regulators (fig 2D) and downregulated in tumors. However, high risk PTCs showed increased expression of this lncRNA. Explain this discrepancy.

In figures 4 significant differences were not so evident. 

Results in paragraph 3.3 were not discussed.

In contrast to what stated at lines 227-229 and 332-334, figure 5 showed that Bendamustine, BTAS-6417, BLU-667 and PD183805 were more sensitive in targeting differentiation regulators in the high-risk group than in low-risk group.

The nomogram was constructed by combining the DR-lncRNAs risk scores with age, not with other predictable clinical factors (line 339). Does the AUC of 3- and 5-year OS of 0.966 and 0.967 refer to figure 6B or 6C? In figure 6C 5-years should be red. In Figure 6D, right panel does “all” line refer to all prognostic factors? “None” line is superfluous. Figure 6D, left panel is not mentioned in the results.

Most of Western blot shown in fig 8E are overexposed. Moreover the interpretation of silencing experiments is contradicting. In particular, depletion of DPH6-DT increased AKT, p-AKT and PI3K expression levels, not decreased as stated at lines 284-285.

Moreover, in the discussion (lines 306-308) the authors stated that downregulation of DPH6-DT inhibits proliferation and metastasis by activating the PI3K-AKT signaling pathway, but the results indicated that downregulation  of DPH6-DT increases proliferation and invasion.
At line 355 the authors stated that silencing DPH6-DT increased the expression of differentiation-related genes. This was not proved. However, since DPH6-DT expression positively correlated with the expression of differentiation genes (fig 2D) and increased with increasing of differentiation degree (fig 7), I would expect the contrary.
Consistently, silencing of DPH6-DT activated PI3K-AKT pathway, that was shown to inhibit the expression of differentiation genes, not to increase it, as stated at line  358.

Minors:
At line 185 correct “elated”

At line 285 correct “was were” 

Author Response

We would like to thank you for your thoughtful comments on our manuscript. We believe that the additional changes we have made in response to your comments have improved the comprehension of the study put forth in the manuscript. Below is our point-by-point response to your comments. We have not listed all minor changes made below. However, revised parts are highlighted in red in the manuscript.

Comment 1:The selection of the initial 1636 lncRNAs should be better explained.

Response:We downloaded the RNA-seq dataset and matched clinical information from University of California Santa Cruz (UCSC) Xena (http://xena.ucsc.edu/). The samples with survival time less than 30 day or incomplete clinical data were excluded from the study. Finally, 492 TC and 58 normal samples from TCGA database were enrolled. Whole-transcriptome sequencing data was performed using FPKM expression level in transcripts per million (TPM). To exclude genes with high variability across patients, we calculated the median absolute deviation (MAD) of the 492 samples. LncRNAs with MAD > 0.5 were defined as genes with high variability and excluded in the RNA-Seq matrix. Meantime, the expression level of lncRNAs was 0 or no clear name annotation also was excluded. Finally, a total of 1636 lncRNAs were enrolled.

Line 167-170 and Figures 2B and 2C: These results are not related with the topic and may be moved in the supplemental data.

Response: According to your nice advice, we moved the figure 2B and 2C into the supplemental data.

Figure 2C and 2D: Explain the meaning of the crosses.

Response: the crosses represents that these gene signatures have no significant relationship (P >0.05).

In contrast to the sentence at lines 173-175, Figure 2D showed a negative correlation of CASC15 and TNRC6C-AS1 with differentiation-related regulators and a positive correlation for the other three lncRNAs. Moreover, all lncRNAs except DPH6-DT were increased in PTCs, therefore I would expect their negative correlation with differentiation-related regulators.

Response: CASC15 and TNRC6C-AS1 were positively related with differentiation-related regulators, whereas other three lncRNAs showed negative correlation, and all lncRNAs except DPH6-DT were increased in PTCs. However, it is inappropriate to speculate that these lncRNAs have negative correlation with differentiation-related regulators due to illogical. Whether the relationship between lncRNAs and differentiation-related regulators does not depend on the differences expression in tumor and normal samples.

Paragraph 3.2: The five prognosis-related DR-lncRNAs were already cited in paragraph 3.1. In Fig. S1 it is not clear their significant correlation with prognosis. What does represent the upper panel of Figure 3B (ratio score/increasing risk score)?

Response: LASSO Cox regression was used to select candidate prognostic makers, we performed LASSO analysis  to further filter prognosis-related DR-lncRNAs (Figure S1), five prognostic elated DR-lncRNAs, including CASC15, LNC00900, AC055720-2, DPH6-DT, and TNRC6C-AS1, showed strong prognostic value. The upper panel of figure 3B represent the distributions of risk scores (based on the DR-lncRNA prognostic signature) of PTC patients in TCGA database.

DPH6-DT expression was positively correlated with thyroid differentiation regulators (fig 2D) and downregulated in tumors. However, high risk PTCs showed increased expression of this lncRNA. Explain this discrepancy.

Response: Thank you very much for your careful review, this is a great question. High risk PTCs are a whole and should be viewed as a whole, rather than individual gene changes. Moreover, we recalculated the coefficients for each lncRNA using R language and draw the same conclusion.

In figures 4 significant differences were not so evident. 

Response: the figure 4 shows the correlation between risk signature and immune cell infiltration. CIBERSORT analysis revealed that the level of eosinophils and activated DCs were significantly higher whereas the level of activated mast cells (MCs), resting MCs, and macrophages M2 were lower in the high-risk group compared with the low-risk group (Figure 4A). Utilizing the ssGSEA algorithm, we found T follicular helper cell, plasmacytoid DC, immature DC, and central memory CD8 T cell were enrichment in low-risk group (Figure 4B). Besides, MCP counter was further confirmed the immune microenvironment and its association with risk score, and revealed the abundance of endothelial cell, neutrophil, and CD8+ T cell were distinctly higher in low-risk group (Figure 4C). 

Results in paragraph 3.3 were not discussed.

Response: the results of paragraph 3.3 was discussed in the section of “Discussion” in line 419-422.

In contrast to what stated at lines 227-229 and 332-334, figure 5 showed that Bendamustine, BTAS-6417, BLU-667 and PD183805 were more sensitive in targeting differentiation regulators in the high-risk group than in low-risk group.

Response: After we careful check, the Bendamustine, BTAS-6417, BLU-667 and PD183805 were more sensitive in targeting differentiation regulators in the low-risk group than in high-risk group.

The nomogram was constructed by combining the DR-lncRNAs risk scores with age, not with other predictable clinical factors (line 339). Does the AUC of 3- and 5-year OS of 0.966 and 0.967 refer to figure 6B or 6C? In figure 6C 5-years should be red. In Figure 6D, right panel does “all” line refer to all prognostic factors? “None” line is superfluous. Figure 6D, left panel is not mentioned in the results.

Response: the nomogram was constructed by combining the DR-lncRNAs risk scores with age. Figure6B: The calibration plots was performed to evaluate the predictive performance of 3- and 5- year OS. Figure 6C: The AUC value of 3- and 5-year OS. In figure 6C,we correct the 5-year OS curve to balck. Figure6D: right panel does “all” line refer to all prognostic factors and “None” line refer to baseline. Figure 6D: left panel represents the clinical impact curve of the nomogram.

Most of Western blot shown in fig 8E are overexposed. Moreover the interpretation of silencing experiments is contradicting. In particular, depletion of DPH6-DT increased AKT, p-AKT and PI3K expression levels, not decreased as stated at lines 284-285.

Response: Thanks for your suggestion. we are sorry for careless mistakes. We have correct the mistake in line 357,  depletion of DPH6-DT increased AKT, p-AKT and PI3K expression levels. Besides, western blot adopts to the original auto exposure mode to ensure the authenticity of the data.

Moreover, in the discussion (lines 306-308) the authors stated that downregulation of DPH6-DT inhibits proliferation and metastasis by activating the PI3K-AKT signaling pathway, but the results indicated that downregulation  of DPH6-DT increases proliferation and invasion.

Response: Thanks for your nice advice. We have correct the mistake in line 385, downregulation of DPH6-DT promotes proliferation and metastasis by activating the PI3K-AKT signaling pathway.

At line 355 the authors stated that silencing DPH6-DT increased the expression of differentiation-related genes. This was not proved. However, since DPH6-DT expression positively correlated with the expression of differentiation genes (fig 2D) and increased with increasing of differentiation degree (fig 7), I would expect the contrary.Consistently, silencing of DPH6-DT activated PI3K-AKT pathway, that was shown to inhibit the expression of differentiation genes, not to increase it, as stated at line 358

Response: I agree with you that it is not proved that silencing DPH6-DT increased the expression of differentiation-related genes, so we delete this sentence.Consistently, silencing of DPH6-DT activated PI3K-AKT pathway to inhibit the expression of differentiation genes, and decreasing the differentiation degree.

We tried our best to improve the manuscript and made some changes in the manuscript.  These changes will not influence the content and framework of the paper. And marked in red in revised paper. We appreciate for your warm work earnestly, and hope that the correction will meet with approval.

Once again, thank you very much for your comments and suggestions.

Round 2

Reviewer 3 Report

In this revised version of the paper the authors did not completely address the criticisms.

1) The selection of initial lncRNA should be also included in the text.

2) I cannot find previous fig.2B and C in supplemental data.

3) In contrast to what stated, Figure 2D showed a negative correlation of CASC15 and TNRC6C-AS1 with differentiation-related regulators (negative R) and a positive correlation for the other three lncRNAs (positive R). Of note, in the responses the authors agreed that DPH6-DT espression was positively correlated with thyroid differentiation, as shown in Fig 2D.

4) In S fig 1 the significant correlation with prognosis of the identified lncRNA is still not clear. Expand the legend and explain what the lines represent. These results should be moved in paragraph 3.1, since the five prognosis-related DR-lncRNAs were already cited at line 222.

5) The authors stated that the relationship between lncRNAs and differentiation-related regulators does not depend on the expression in tumor. Nevertheless, if a lncRNA is positively associated with thyroid differentiation I would expect a lower espression in tumors, which are associated with lower expression of thyroid differentiation genes, as for DPH6-DT, and viceversa. This should be commented in the Discussion.

6) In figures 4 significant differences were still not evident. May the authors provide row data?

7) The colors of high and low risk tumors in fig. 5b should be uniformed.

8) The nomogram was constructed by combining the DR-lncRNAs risk scores with age, not with other predictable clinical factors. Correct at line 421.

9) Figure 6D left panel is still not mentioned in the results.

10) The original auto exposure mode is not always the best one. A manual and more adequate exposure would not compromise the authenticity of the data of western blot.

10) The authors agreed that silencing of DPH6-DT activated PI3K-AKT pathway to inhibit the expression of differentiation genes. However in the Discussion they stated that PI3K-AKT pathway increases the expression of NIS in PTC. This Is not true. Moreover, in fig 4 (not citred in the text) the role of AKT in differentiation (or dedifferentiation) is ambiguous

Author Response

Dear Reviewer

    Thank you very much for your suggestions and comments, and we appreciate for your warm work earnestly. Revised parts are highlighted in red in the manuscript. The major corrections in the manuscript and the point-by point replies are as follows: 

  • The selection of initial lncRNA should be also included in the text.

Response: According to your suggestion, the selection of initial lncRNA was included in line 102-107

  • I cannot find previous fig.2B and C in supplemental data.

Response: We feel sorry for our carelessness, we put the previous fig.2B and C into   supplemental data (fig.S1A and B). 

  • In contrast to what stated, Figure 2D showed a negative correlation of CASC15 and TNRC6C-AS1 with differentiation-related regulators (negative R) and a positive correlation for the other three lncRNAs (positive R). Of note, in the responses the authors agreed that DPH6-DT espression was positively correlated with thyroid differentiation, as shown in Fig 2D.

Response: Sincerest thanks for your comments. After we carefully check, Figure 2D showed that CASC15 and TNRC6C-AS1 were negatively related with differentiation-related regulators, whereas other three lncRNAs showed positive correlation (line 239-240).

  • In S fig 1 the significant correlation with prognosis of the identified lncRNA is still not clear. Expand the legend and explain what the lines represent. These results should be moved in paragraph 3.1, since the five prognosis-related DR-lncRNAs were already cited at line 222.

Response: According to your nice suggestion, we move the results of fig S2 into the paragraph 3.1. Fig.S2A: Lasso regression of the 5 DR-lncRNAs, the lines represent the DR-lncRNA. S2B: Cross-validation for tuning the parameter selection in the LASSO regression (λ=5) (line 474-475). note: The least absolute shrinkage and selection operator (LASSO) Cox regression was used to select candidate prognostic DR-lncRNAs. Penalty parameter (λ) for the model was determined by 10 fold cross-validation following the minimum criteria (i.e. the value of λ corresponding to the lowest partial likelihood deviance). We hope this annotation helps explain the confusion.

  • The authors stated that the relationship between lncRNAs and differentiation-related regulators does not depend on the expression in tumor. Nevertheless, if a lncRNA is positively associated with thyroid differentiation I would expect a lower expression in tumors, which are associated with lower expression of thyroid differentiation genes, as for DPH6-DT, and viceversa. This should be commented in the Discussion.

Response: Thank you for your constructive comments, we discussed this parts in line 456-461.

  • In figures 4 significant differences were still not evident. May the authors provide row data?

Response: we used the following to code run our data again, we also invited a professional to run our data by R software, and still draw the same conclusion. We provide all row data in the supplementary data. The code is as follows:

#ssGSEA

ssgsea=read.csv("ssGSEA result.csv",header = T,row.names = 1)

ssgsea=ssgsea[,match(rownames(riskscoretable),colnames(ssgsea))]

ssgseaboxdata=data.frame(t(ssgsea),check.names = F)

ssgseaboxdata$Risk=riskscoretable$risk

ssgseabox = gather(ssgseaboxdata, CellType, Composition,  -Risk)

ggboxplot(ssgseabox,

          x="CellType",

          y="Composition",

          color = "Risk",palette = "aaas",

          add = "jitter",

          xlab = "",title = "ssGSEA",

          ylab = "ssGSEA score") +

  theme_get() +

  stat_compare_means(aes(group = Risk),label = "p.signif", hide.ns = T) +

  theme(axis.text.x = element_text(angle = 45,hjust = 1,vjust = 1))

  • The colors of high and low risk tumors in fig. 5b should be uniformed.

Response: the fig.5B was uniformed

  • The nomogram was constructed by combining the DR-lncRNAs risk scores with age, not with other predictable clinical factors. Correct at line 421.

Response: I agree with you, and correct the sentence according to your advice.

  • Figure 6D left panel is still not mentioned in the results.

Response: Thanks for your reminding, the results was described in line 308-310

  • The original auto exposure mode is not always the best one. A manual and more adequate exposure would not compromise the authenticity of the data of western blot.

Response: according to your suggestion, we performed the western blotting assay again to obtain well pictures. If you are not satisfied, we can continue to repeat the experiment. Thank you again.

  • The authors agreed that silencing of DPH6-DT activated PI3K-AKT pathway to inhibit the expression of differentiation genes. However in the Discussion they stated that PI3K-AKT pathway increases the expression of NIS in PTC. This Is not true. Moreover, in fig 4 (not citred in the text) the role of AKT in differentiation (or dedifferentiation) is ambiguous

Response: Thanks for your reminding, The PI3K-AKT pathway has been well recognized in the regulation of cell differentiation via decreasing the expression of NIS in PTC. The fig4 was citred in line 399. The expression of AKT was negative with DPH6-DT, indicating the role of AKT is dedifferentiation.

Round 3

Reviewer 3 Report

In this revised version the authors included some of the replies in the Supplemental material, however in Supplemental file I found only Table S1.

Why did the authors remove the results relative to BLU-667 and PD183805?

The results of Fig 6D reported in the text are only relative to right panel, delete the left panel.

I cannot find the citation of Fig. 9 in the text. Correct in Fig 9 "defferentiation" with "dedifferentiation"

Author Response

Dear Reviewer

Thank you very much for your suggestions and comments, and we appreciate for your warm work earnestly.

Comment 1:In this revised version the authors included some of the replies in the Supplemental material, however in Supplemental file I found only Table S1.

Response:Thanks for your patients, we uploaded table S1 again.

Comment 2:Why did the authors remove the results relative to BLU-667 and PD183805?

Response:After we carefully calculated, the BLU-667 and PD183805 did not meet the threshold criteria of Pearson’s coefficient > 0.6 and p value <0.01.

Comment 3:The results of Fig 6D reported in the text are only relative to right panel, delete the left panel.

Response:According your nice suggestion, we delete the left panel.

Comment 4:I cannot find the citation of Fig. 9 in the text. Correct in Fig 9 "defferentiation" with "dedifferentiation"

Response:Fig.9 was cited in lines 372. We have carefully checked the manuscript and corrected the errors accordingly.

Thanks for your warm work again! If you have any queries, please don’t hesitate to contact me.